# A Holistic Approach from Systems Biology Reveals the Direct Influence of the Quorum-Sensing Phenomenon on *Pseudomonas aeruginosa* Metabolism to Pyoverdine Biosynthesis

**DOI:** 10.3390/metabo13050659

**Published:** 2023-05-16

**Authors:** Diana Carolina Clavijo-Buriticá, Catalina Arévalo-Ferro, Andrés Fernando González Barrios

**Affiliations:** 1Grupo de Comunicación y Comunidades Bacterianas, Departamento de Biología, Universidad Nacional de Colombia, Carrera 45 No. 26-85, Bogotá 111321, Colombia; carevalof@unal.edu.co; 2Grupo de Diseño de Productos y Procesos (GDPP), Departamento de Ingeniería Química y de Alimentos, Universidad de los Andes, Edificio Mario Laserna, Carrera 1 Este No. 19ª-40, Bogotá 111711, Colombia; andgonza@uniandes.edu.co

**Keywords:** biological network reconstruction, biological network modeling, multiscale models, flux balance analysis, quorum-sensing, pyoverdine, *Pseudomonas aeruginosa*

## Abstract

Computational modeling and simulation of biological systems have become valuable tools for understanding and predicting cellular performance and phenotype generation. This work aimed to construct, model, and dynamically simulate the virulence factor pyoverdine (PVD) biosynthesis in *Pseudomonas aeruginosa* through a systemic approach, considering that the metabolic pathway of PVD synthesis is regulated by the quorum-sensing (QS) phenomenon. The methodology comprised three main stages: (i) Construction, modeling, and validation of the QS gene regulatory network that controls PVD synthesis in *P. aeruginosa* strain PAO1; (ii) construction, curating, and modeling of the metabolic network of *P. aeruginosa* using the flux balance analysis (FBA) approach; (iii) integration and modeling of these two networks into an integrative model using the dynamic flux balance analysis (DFBA) approximation, followed, finally, by an in vitro validation of the integrated model for PVD synthesis in *P. aeruginosa* as a function of QS signaling. The QS gene network, constructed using the standard System Biology Markup Language, comprised 114 chemical species and 103 reactions and was modeled as a deterministic system following the kinetic based on mass action law. This model showed that the higher the bacterial growth, the higher the extracellular concentration of QS signal molecules, thus emulating the natural behavior of *P. aeruginosa* PAO1. The *P. aeruginosa* metabolic network model was constructed based on the iMO1056 model, the *P. aeruginosa* PAO1 strain genomic annotation, and the metabolic pathway of PVD synthesis. The metabolic network model included the PVD synthesis, transport, exchange reactions, and the QS signal molecules. This metabolic network model was curated and then modeled under the FBA approximation, using biomass maximization as the objective function (optimization problem, a term borrowed from the engineering field). Next, chemical reactions shared by both network models were chosen to combine them into an integrative model. To this end, the fluxes of these reactions, obtained from the QS network model, were fixed in the metabolic network model as constraints of the optimization problem using the DFBA approximation. Finally, simulations of the integrative model (CCBM1146, comprising 1123 reactions and 880 metabolites) were run using the DFBA approximation to get (i) the flux profile for each reaction, (ii) the bacterial growth profile, (iii) the biomass profile, and (iv) the concentration profiles of metabolites of interest such as glucose, PVD, and QS signal molecules. The CCBM1146 model showed that the QS phenomenon directly influences the *P. aeruginosa* metabolism to PVD biosynthesis as a function of the change in QS signal intensity. The CCBM1146 model made it possible to characterize and explain the complex and emergent behavior generated by the interactions between the two networks, which would have been impossible to do by studying each system’s individual components or scales separately. This work is the first in silico report of an integrative model comprising the QS gene regulatory network and the metabolic network of *P. aeruginosa*.

## 1. Introduction

In recent years, computational modeling of biological systems has acquired a significant role in understanding and predicting cellular performance. Computational modeling for explaining cell behavior has been accomplished by numerous discrete, continuous, and spatially explicit models describing subcellular systems and processes that biological systems need to survive [1]. Indeed, all biological systems are complex and dynamic and can be modeled as clusters of various functional networks. Metabolic, cell signaling, gene regulation, and intercellular communication networks are interconnected, regulate each other, and operate in diverse temporal and spatial domains to maintain living organisms’ growth, development, and reproductive potential [2]. The functional networks of biological systems can operate on characteristic and specific temporal and spatial scales. For instance, intracellular biochemical reactions can ensue on time scales of seconds or less, involving gradients exceeding microns in length. Functional phenotypes at the cellular and cell-to-cell levels unfold in tens of seconds to minutes and on spatial scales of tens of microns [3]. Therefore, biological systems are inherently considered as hierarchical and multiscale in time and space. Most traditional computational models focus on studying biological systems at single biological scales. However, systemic approaches facilitate the designing of multiscale computational models, to bridge those scales, so that the effects of perturbations at one scale can be predicted at other scales. Thus, multiscale computational models are exceptionally positioned to capture the relationship between different scales of biological function; they can bridge the gap between isolated in vitro experiments and whole-organism in vivo models [1,2,3].

Following the development of next-generation sequencing techniques, a key objective has been to relate annotated genome sequences to cellular physiological functions. Therefore, systems biology has been designing new computational strategies to reconstruct biological networks (genomic and metabolic) and to model and dynamically simulate biological systems in order to study, from a holistic perspective, the regulation of mechanisms leading to the expression of diverse phenotypes. Thus, the extensive dataset of measurements of all biomolecules at the system level favors integration from the molecular to the whole organism level. In recent decades, several computational studies have made developing multiscale models of some biological systems possible. For example, Thiele et al. and Biggs and Papin have shown the advantages of multiscale models for understanding metabolic and macromolecular synthesis in *Escherichia coli* and biofilm formation by *Pseudomonas aeruginosa*, respectively [4,5].

Following a holistic approach, this work aimed to explain the synthesis of the siderophore pyoverdine (PVD), a virulence factor in *P. aeruginosa* [6]. The PVD synthesis requires two biological processes. The first is the expression of *pvd* genes, which encode enzymes involved in PVD synthesis and which are regulated by the quorum-sensing (QS) phenomenon [7] (a mechanism of cell communication mediated by signaling molecules *-in this case, induced under conditions of iron deficiency*, and responsible for synchronizing phenotype expression in a bacterial community—*in this case, the factor virulence PVD expression*) [8]. The second process corresponds to the metabolic pathway involved in PVD synthesis. Therefore, to understand how these two biological processes work and how they regulate each other, it was necessary to design a methodology to develop a model capable of describing their interactions.

*P. aeruginosa* has two QS systems, LasI/LasR and RhlI/RhIR. The signaling molecule of the LasI/LasR system is N-(3-Oxododecanoyl)-L-homoserine lactone (3O-C12-HSL), which is synthesized by LasI and detected by the LasR receptor. The RhlI/RhIR system is regulated by N-butanoyl-L-homoserine lactone (C4-HSL), synthesized by RhII, and detected by the RhIR receptor. Both QS systems are hierarchically organized, such that the LasI/LasR system regulates the function of the RhlI/RhIR system, allowing further fine-tuning of the QS feedback on specific genes [9]. These QS systems are linked through regulatory connections involving regulatory proteins (RsaL, PqsR, RclA, vfR, and GacA) and the *Pseudomonas Quinolone Signal* (PQS) system. This system encodes the synthesis of 2-heptyl-3-hydroxy-4-quinolone (2C7-3OH-4(1H)Q) that forms a complex with the available ferric ion (Fe^3+^) and plays an essential role in the *pvd* genes expression. At the cellular level, the quinolone complex positively controls the expression of the Pvd-type proteins, which is responsible for PVD maturation, synthesis, and exportation [10].

Most PVD biosynthesis is carried out in the bacterial cytosol by four non-ribosomal peptide synthase (NRPS) proteins encoded by the *pvdL*, *pvdI*, *pvdJ*, and *pvdD* genes [6]. Together, these genes encode ferribactin synthase, responsible for synthesizing ferribactin, the precursor of PVD. PVD biosynthesis also requires the expression of the *pvdA*, *pvdF*, and *pvdH* genes. According to the model proposed by Visca et al. [8], the PvdE protein (an ABC-type transporter), located in the inner membrane, transports ferribactin to the periplasmic space where PVD chromophore maturation occurs by an oxidation-reduction mechanism driven by the PvdM, PvdP, PvdQ, PvdO, and PvdN proteins [7,11,12] through a mechanism not fully elucidated. Finally, mature PVD is secreted into the extracellular space by an unclear process that chelates iron and forms the PVD-Fe^3+^ complex, which is transported back into the intracellular space by the FpvA carrier. Iron is then released from PVD to the periplasm by a mechanism involving iron reduction (Fe^3+^ to Fe^2+^), and PVD is then transported out of the cell via a specialized carrier of the siderophore recycling process [13,14,15,16].

An integrative (multiscale) model that combines the QS gene regulatory network and the metabolic network of *P. aeruginosa* is proposed here for the first time to describe and understand the influence of the QS phenomenon on PVD biosynthesis in *P. aeruginosa* from a systemic perspective. That model involves the process of bacterial communication mediated by the QS phenomenon for synthesizing signal molecules that regulate the expression of *pvd* genes to generate enzymes that metabolically catalyze the production of the virulence factor. Furthermore, this integrative model combined two biological network models: (i) The QS gene regulatory network model, based on deterministic approaches and whose simulation results were the concentration of chemical species and the fluxes of chemical reactions; and (ii) the model of the *P. aeruginosa* metabolic network based on the flux balance analysis (FBA) approximation. This approximation addressed the biomass maximization problem constrained by mass balance and thermodynamic conditions to obtain the optimal distributions of reaction fluxes through the metabolic network [17,18,19,20,21]. The methodological strategy first identified reactions shared by the QS and the *P. aeruginosa* metabolic networks. Then, the fluxes of these shared reactions, obtained from the QS network, were used as a constraint system to model the *P. aeruginosa* metabolic network by dynamic flux balance analysis (DFBA) [22,23]. The proposed multiscale model showed a capability to infer the influence of QS on the *P. aeruginosa* strain PAO1 metabolism to PVD biosynthesis as a function of variations in QS signal intensity.

## 2. Materials and Methods

Computational modeling followed three steps: (i) construction of a deterministic model of the *P. aeruginosa* QS network as a gene regulatory network for the expression of *pvd* genes encoding enzymes for PVD synthesis and maturation; (ii) construction of the *P. aeruginosa* genome-scale metabolic network, including PVD biosynthetic reactions using the FBA approximation; and (iii) integration of these two network models into a multiscale model using the DFBA approximation (Figure 1).

### 2.1. Construction and Modeling of the Quorum-Sensing Gene Regulatory Network

The following elements were used to construct and model the QS gene regulatory network, (i) bibliomic information about the genes involved in the QS process, (ii) a language/platform for the biological representation of the retrieved bibliomic data, and (iii) a language/platform for the mathematical modeling of the biological representation.

Bibliomic information was retrieved from generic and specialized databases, such as the Scopus and Web of Science bibliographic indexes and specific biological databases. In addition, several search queries were used to find relevant documents and database entries that provided the maximum number of chemical species for modeling the biological process.

The QS network model was built using the standard System Biology Markup Language (SBML) format. The CellDesigner 4.4 software [24] was used to model the biological processes by applying the Systems Biology Graphical Notation (SBGN) and to create the mathematical representation of the biological model with a MathML layer. The resulting model was based on bibliomic analysis and information stored in the GenomeNet (https://www.genome.jp, accessed on 5 March 2016), KEGG [25,26], BioCyc [27,28], UniProtKB/Swiss-Prot [29] and PubChem [30,31] databases, as well as on the search of annotated genes and their corresponding mRNAs and proteins to manually curate the model. Based on the biological model, a system of 114 ordinary differential equations (ODEs) representing the interactions between chemical species was established in a manually curated QS network model (Figure 2). 

For QS modeling, the initial concentrations of genes, QS regulatory proteins, and available cytosolic Fe^3+^ were fixed (data available in Mendeley Data, https://doi.org/10.17632/2xzzkmnpfx.1). In addition, each reaction incorporated the kinetic constant (k) values (a total of 103 kinetic parameters data are available in Mendeley Data, https://doi.org/10.17632/2xzzkmnpfx.1) and the links of each interaction between related molecules. The corresponding ODEs for mRNAs and proteins in the transcription and translation processes are exemplified in Equations (1) and (2), respectively [32].
(1)$$\frac{dYmRNA}{\mathrm{dt}}={\;k}_{a\;}\left[Ygene\right]-{\;k}_{b\;}\left[YmRNA\right]-k_{c\;}\left[YmRNA\right]$$
(2)$$\frac{\mathrm{dYProtein}}{\mathrm{dt}}={\;k}_{1\;}\left[YmRNA\right]-{\;k}_{2\;}\left[\mathrm{YProtein}\right]-k_{3}\left[\mathrm{YProtein}\right]$$
where Y represents the chemical species, $k_{a\;}\;$the transcription rate constant, $k_{b\;}$and $k_{1}$ the translation rate constants, $k_{c\;}$and $k_{3\;}$the degradation rate constants, and $k_{2\;}$the consumption rate constant of each reaction in the system. 

To construct the deterministic QS model, elementary kinetics, based on the law of mass action for all chemical species, was assumed for all reactions in the system. QS model simulations were run using SOSlib [33] in the CellDesigner software, which solves the rigidity problem of the ODEs and initiates numerical integration using the backward differentiation formula (BDF) or the Adams-Moulton (AM) method to calculate *x*(*t*) for a series of time points [34].

#### Simulation Scenario Conditions for the Quorum-Sensing Network

After testing the stability of the QS network by the behavior of signal molecules, intracellular ferribactin production, and intracellular and extracellular PVD production, a single condition was set for the different simulation scenarios, namely, the initial concentration of the extracellular signal molecule PQS (E-PQS). This ranged from 0.01 μM to 0.1 μM with 0.01-unit intervals and from 0.01 μM to 0.1 μM with 0.1-unit intervals for a total of 20 simulation scenarios (Table 1). These simulation scenarios were performed to emulate the behavior of in vitro cultures of *P. aeruginosa*. When the microbial population density increases in these cultures, the concentration of QS signal molecules that regulate the expression of bacterial phenotypes, such as PVD, also increases [9,35,36]. In addition, the simulations gave results for changing metabolite concentrations and reaction fluxes in time $\left({\mu \mathrm{mol}\;s}^{-1}L^{-1}\right)$. The latter ones were used in the subsequent modeling of the *P. aeruginosa* metabolic network.

### 2.2. Construction of the Pseudomonas aeruginosa Metabolic Network 

The in silico construction of the *P. aeruginosa* metabolic network required a previously published generic cell model of the bacterium, the specific metabolic pathway of PVD synthesis, and software for modeling the hundreds of reaction equations representing the metabolic pathways. 

The computational modeling was based on three data sources: (i) The genome-scale metabolic network model iMO1056 [37], which was to be used as a template, because it was the first genome-scale metabolic network construction of *P. aeruginosa* published in the literature; (ii) the *P. aeruginosa* genome annotation in the PseudoCAP database; and (iii) the metabolic pathway of PVD synthesis in the MetaCyc database (PseudoCyc), which was extended by bibliomics. As in the IMO1056 model, each reaction was manually mapped against data in biological databases (KEGG, BioCyc [27], ModelSEED [38], MetanetX [39], TransportDB [40], TCDB [41]) for preliminary curation. This process updated and complemented the network regarding stoichiometry, directionality, biological evidence, and subcellular location of each reaction, as well as in the gene-protein-reaction (GPR) associations. Furthermore, specific reactions involved in the PVD metabolic synthesis, some exchange reactions involving QS signal molecules, reactions for PVD transport, and reactions in bacterial culture (Luria-Bertani, LB) medium, were considered for computational modeling (data available in Mendeley Data, https://doi.org/10.17632/y9htx3fcjm.1).

#### 2.2.1. Curation of the *Pseudomonas aeruginosa* Metabolic Network

Curing a metabolic process refers to making the network mathematically feasible by removing pathologies such as metabolites that are not produced or consumed by any reaction in the model, eliminating thermodynamically infeasible cycles, and validating the production of metabolites in the biomass reaction.

Before curating the *P. aeruginosa* metabolic network, the list of reactions in the network was represented as a stoichiometric matrix $(S_{ij}$) containing the ratio of the stoichiometric coefficient of each metabolite (i) in the reaction (j). Next, this network was curated according to the following steps: (i) validation of the production of the biomass precursors by reverse engineering [42], (ii) addition of the PVD metabolite to the biomass reaction by increasing the value of the PVD coefficient until the metabolic pathway of PVD synthesis was activated, (iii) detection and solution of network pathologies such as root no-production and root no-consumption metabolites [43,44], and (iv) search and solution of thermodynamically infeasible cycles (TICs) in the network [45]. In this work, the proposal to add PVD to the biomass reaction was inspired by the work of Amara et al. [46], Prigent et al. [47], and Kim et al. [48], in which efforts have been made to optimize the production of secondary metabolites, such as virulence factors, in genome-scale metabolic models.

#### 2.2.2. Modeling of the *Pseudomonas aeruginosa* Metabolic Network Using a Steady-State FBA Approximation

The *P. aeruginosa* metabolic network modeling using a steady-state FBA approximation [17] comprised the representation of a system of differential equations coupled to an objective function, i.e., the biomass maximization (an optimization problem, a term borrowed from engineering). The optimization problem was formulated as described in Equation (3) and solved as a linear programming (LP) problem using the CPLEX solver in the GAMS software (General Algebraic Modeling System: https://www.gams.com/, accessed on 20 March 2023).
$$Maximize\;\;\mu =v_{j\left(Biomass\right)\;}$$
subject to:$$\overset{J}{\underset{j=1}{\sum }}\;S_{ij}*\;v_{j}=0,\;\;\;\;\;\forall \;i=1,\ldots ,I$$
(3)$$LB_{j}\leq \;v_{j}\leq UB_{j},\;\;\;\;\;\forall \;\;j=1,\ldots ,J\;j^{R}\;j^{IR}\in J$$
$$v_{O2}=K_{1}$$
$$v_{Glc}=K_{2}$$
$$v_{ATP}=K_{3}$$
where μ is the objective function, I  and J  represent the total number of metabolites and reactions, respectively; $\;S_{ij}$ is the stoichiometric matrix for each metabolite i in the reaction j; and $v_{j}$ is the flux of the reaction j expressed in ${\mathrm{mmol}\;\mathrm{gDW}}^{-1}h^{-1};\mathrm{and}\;LB_{j}$ and $UB_{j}$ are the minimum and maximum fluxes that can adopt every reaction j  given by the thermodynamic feasibility of reactions. The fluxes for oxygen ($v_{O2}$) and glucose ($v_{Glc}$) uptake, and ATP ($v_{ATP}$) production were limited at the rates of $-10{\;\mathrm{mmol}\;\mathrm{gDW}}^{-1}h^{-1}\;$($K_{1}$ and $K_{2}$), and $10\;{\mathrm{mmol}\;\mathrm{gDW}}^{-1}h^{-1}\;$($K_{3}$) [37].

### 2.3. Combining the QS Gene Regulatory Network and the Pseudomonas aeruginosa Metabolic Network Models into an Integrative Model

The QS gene regulatory network and the *P. aeruginosa* metabolic network represent two types of biological processes; both are part of a living cell but differ in time scale, spatial location, control mechanisms, and chemical species that usually work separately. However, both networks share chemical species, such as the enzymes encoded by the QS network genes that catalyze some biochemical reactions in the *P. aeruginosa* metabolic network. The challenge was to combine these two networks to function as an integrated multiscale model. The work of Mallmann and collaborators inspired the methodological proposal to solve the challenge [49]. Thus, the challenge was solved by (i) Finding the shared reactions between the two models: the QS gene regulatory network model and *Pseudomonas aeruginosa* metabolic network model, and (ii) using the fluxes of the reactions shared by both model as constraints in the multi-stage FBA (see Section 2.3.2) and DFBA (see Section 2.3.3) simulations. The two networks were found to share nine reactions involved in the synthesis and transport of QS signal molecules (PQS, 3O-C12-HSL, and C4-HSL), ferribactin and PVD (Table 2). 

#### 2.3.1. Design of Simulation Scenarios

The integrated model was evaluated through several simulated scenarios expecting its results to match the known cellular behavior of an in vitro bacterial culture of *P. aeruginosa* [50,51]. For the simulations, six scenarios were selected for subsequent simulations using the multi-stage FBA and DFBA approximations. Scenarios Sc1, Sc2, and Sc3 included reactions involved in the synthesis of QS signal molecules and those involved in ferribactin synthesis (Sc1), PVD synthesis (Sc2), or PVD transport (Sc3). Scenarios Sc4, Sc5, and Sc6 included reactions involved in the transport of QS signal molecules and those involved in ferribactin synthesis (Sc4), PVD synthesis (Sc5), or PVD transport (Sc6) (Table 3). These simulation scenarios were performed to evaluate which combination of reactions in the multi-stage FBA (see Section 2.3.2) and DFBA (see Section 2.3.3) simulations showed a more significant change in the fluxes’ distribution. Scenario 6 (Sc6) was chosen.

#### 2.3.2. Simulation Using the Multi-Stage FBA Approximation

Under the multi-stage FBA approximation, both network models were combined by fixing in the metabolic model, as equality constraints, the fluxes of the shared reactions, obtained from the QS network simulations according to each simulation scenario where E-PQS concentration was changed (Table 1). The units of the fluxes of the QS reactions obtained from the QS network simulations (${\mu \mathrm{mol}\;s}^{-1}L^{-1}$) were adjusted to ${\mathrm{mmol}\;\mathrm{gDW}}^{-1}h^{-1}$ [52]. Simulations were run at a time scale of 2000 intervals in the six simulation scenarios (Table 3). The flux value obtained from the QS model in each time interval of the shared reactions, according to the reactions involved in the different scenarios (Table 3), was fixed respectively for each time interval in the multi-stage FBA and DFBA (see Section 2.3.3) simulations. In addition, the optimization problem (Equation (3)) was modified for the multi-stage FBA (Equation (4)) and solved as a linear programming (LP) problem using the CPLEX v.12.6.0.0 solver in GAMS (https://www.gams.com/latest/docs/S_CPLEX.html, accessed on 20 March 2023).
$$Maximize\;\;\mu =v_{j\left(Biomass\right)\;}$$

Subject to:$$\overset{J}{\underset{j=1}{\sum }}\;S_{ij}*\;v_{j}=0,\;\;\;\;\;\;\forall \;i=1,\ldots ,I\;\;\;\;\;\;\;$$
(4)$$LB_{j}\leq \;v_{j}\leq UB,\;\;\;\;\;\;\forall \;\;j=1,\ldots ,J\;j^{R}\;j^{IR}\in J\;\;\;\;\;\;\;\;\;\;\;\;$$
$$v_{O2}=K_{1}\;\;\;\;\;$$
$$v_{Glc}=K_{2}\;\;\;\;\;$$
$$v_{ATP}\;=K_{3}\;\;\;\;\;$$
$$v_{j}=v_{\hat{j}}\;\;\;\;\;\forall \;\;\hat{j}\in QS\;Reactions$$
$$\forall \;t\in \left[t_{0},\;t_{f}\right]$$
where $\hat{j}\;$ is the subset of J-type reactions denoting the reactions shared by the *QS* and *P. aeruginosa* metabolic networks, w is the instance to evaluate μ in $\hat{j}$-type reactions, α is the final instance to evaluate μ in $\hat{j}$-type reactions and β is the last $\hat{j}$-type reaction to be fixed. For this model, the fluxes of the shared reactions ($v_{\hat{j}}$) were fixed according to each simulation scenario (Table 3). Finally, scenario 6 (Sc6) showed a more significant change in the fluxes’ distribution.

#### 2.3.3. Simulation Using the DFBA Approximation

The behavior of the integrated model over time was observed by dynamic simulation using the DFBA approximation. Thus, observing the influence of the QS network on the *P. aeruginosa* metabolic network and the interaction between both networks over time was possible. Furthermore, the DFBA approximation proposed by Mahadevan et al. [22] was used to solve the optimization problem shown in Equation (5).
$$Maximize\overset{N}{\underset{j=1}{\sum }}c_{j}\cdot v_{j}\left(t\right)$$

Subject to:$$z_{i}\left(t+\Delta t\right)=z_{i}\left(t\right)-\overset{J}{\underset{j=1}{\sum }}S_{ij}\cdot v_{j}\left(t\right)\cdot X\left(t\right)\cdot \Delta t\;\;\;\;\;\forall \;i\;\epsilon \;1,\ldots ,\;I_{extracellular}$$
(5)$$X\left(t+\Delta t\right)=X\left(t\right)+µ\cdot X\left(t\right)\cdot \Delta t$$
$$\overset{J}{\underset{j=1}{\sum }}S_{ij}\cdot v_{j}=0\;\;\;\;\;\forall \;i\;\epsilon \;1,\ldots ,\;I_{extracellular}$$
$$v_{j}^{min}<v_{j}\left(t\right)<v_{j}^{max\;}\;\;\;\;\;\forall \;j\;\epsilon \;1,\ldots ,\;J$$
$$\hat{c}\left(z_{i}\left(t\right),\;v_{j}\left(t\right)\right)\leq 0\;\;\;\;\;\;\;\;\;\;\;\;\;\;\;\;\;\;\;\;\;\;\;\;\;\;\;\;\;\;\;\;\;\;\;\;$$
$$z_{i}\left(t\right)\geq 0\;\;\;\;\;\;\;z_{i}\left(t_{0}\right)=z_{i,0}\;\;\;\;\;\;\forall \;i\;\epsilon \;1,\ldots \;,\;I_{extracellular}$$
$$X\left(t\right)\geq 0\;\;\;\;\;X\left(t_{0}\right)=X_{0}\;\;\;\;\;\;\;\;\;\;\;\;\;\;\;\;\;\;\;\;\;\;\;\;\;\;\;\;\;\;\;\;\;\;$$
$$\Delta t=\frac{t_{f}-t_{0}}{G}\;\;\;\;\;\;\forall \;G\;\epsilon \;0\ldots G$$
$$v_{j}=v_{\hat{j}}\;\;\;\;\;\forall \;\hat{j}\;\epsilon \;Qs\;Reactions$$
$$\forall \;t\;\epsilon \;\left[t_{0},\;t_{f}\right]\;\;\;$$
where $z_{i}$ is the extracellular metabolite concentration, $z_{i,0}$ and $X_{0}\;$ are the initial conditions for each metabolite and biomass concentration, respectively, µ is the specific cell growth rate, $c_{j}$ is the reaction weight, $\hat{c}\left(z,v\right)$ is the vector of nonlinear constraints (substrate consumption kinetics), $t_{0}$ and $t_{f}$ correspond to the initial and final times (0 and 19 h, respectively, at 0.0095-h intervals), G is the number of intervals used to discretize the time and $\hat{j}\;$ is the subset of J-type reactions denoting the reactions shared by the *QS* and *P. aeruginosa* metabolic networks. Simulations were run to obtain (i) the flux profile of each reaction, (ii) the growth rate profile, (iii) the biomass concentration profile, and (iv) the concentration profiles over time of the metabolites of interest, i.e., the QS signal molecules, glucose, and PVD. The optimization problem was solved as a nonlinear programming problem (NLP) using the CONOPT v3.15N solver in GAMS software (https://www.gams.com/latest/docs/S_CONOPT.html, accessed on 20 March 2023). This model includes the equality and inequality constraints of the system (thermodynamics) and the fluxes of the QS network reactions fixed in the *P. aeruginosa* metabolic network (according to scenarios Table 3) as equality constraints according to the results of each simulation scenario where E-PQS concentration was changed (Table 1). The following conditions were set for the simulation: 0.0095-time step size and three orthogonal placement points for a Legendre polynomial; also, initial source concentrations of carbon (glucose = 55.5 mM), nitrogen (NH_4_^+^ = 104.127 mM), phosphate (Pi = 3.88 mM), sulfate (SO_4_^2−^ = 0.286 mM), biomass (X = 0.1 g/L) and oxygen (O_2_ = 1 mM). The initial concentrations of glucose, nitrogen, phosphate, and sulfate were calculated according to the approximate composition of the Luria Broth (LB) culture medium reported in the technical specifications of the commercial culture medium.

### 2.4. Cultures of Pseudomonas aeruginosa Strain PAO1

The integrated model, named CCBM1146, was evaluated qualitatively by comparing the behavior pattern of PVD production and biomass production between data obtained from in silico simulations and those obtained from in vitro cultures of *P. aeruginosa* strain PAO1 (collection of the research group *Comunicación y comunidades bacterianas*, Department of Biology, Faculty of Sciences, Universidad Nacional de Colombia, Bogotá) was grown in Luria Broth medium (10 g/L tryptone, 5 g/L yeast extract, 10 g/L NaCl) in a 1-L BioStat^®^A bioreactor (Sartorius Stedim Biotech, Goettingen, Germany), equipped with a disc-type impeller, a ring-type diffused bubble aeration system, pH and pO_2_ probes, a chiller for temperature control, and an aeration system for continuous and automatic control of the laboratory airflow. Cultures were grown in triplicate batches under controlled pH = 7, Tº = 37 °C, agitation = 200 rpm, oxygen saturation = 20%, and air flow = 2 L*min^−1^ (2 vvm). Cultures were incubated for 24 h, and aliquots were collected at 1-h intervals.

#### 2.4.1. Evaluation of Bacterial Growth

Aliquots of 1 mL of the *P. aeruginosa* cultures were taken to measure the absorbance at 600 nm by spectrophotometry. The mean values were analyzed by logistic regression using the StatSoft statistical software to fit the data according to Equation (6) and obtain the following specific growth equation for strain PAO1.
(6)$$v2=Exp\;{\left(\frac{\left(-a+b\right)*v1}{1+Exp\left(-a+\left(b\right)*v1\right)}\right)}_{}$$
where v2 corresponds to absorbance and v1 to time. Values calculated for a (1.950216) and b (0.234159), showed an adequate adjustment of the data with R=0.9915 and explained variance=98.316%.

#### 2.4.2. Evaluation of Biomass Production

After incubation was complete, three aliquots (1 mL) of the in vitro cultures of *P. aeruginosa* were taken and centrifuged at 14,000 rpm for 5 m at 4 °C. The pellets were discarded, and the supernatants were stored at −80 °C for subsequent determination of the PVD profile following the protocol of Meyer and Abdallah [53,54,55]. Total biomass was measured as dry weight. Briefly, at the end of the culture incubation, three aliquots (50 mL) of the cultures were collected and centrifuged at 14,000 rpm for 5 m. The pellets were washed with sterile water, centrifuged again at 14,000 rpm for 5 m, dried at 50 °C, transferred to a drying chamber, and weighed.

## 3. Results

### 3.1. The Quorum-Sensing Gene Regulatory Network for Pyoverdine Expression in Pseudomonas aeruginosa: A Deterministic Model

The QS network model described here consists of 114 subcellular chemical species, including proteins, small molecules, genes and their corresponding mRNAs, and nine complexes containing different molecules (data available in Mendeley Data, https://doi.org/10.17632/2xzzkmnpfx.1). In addition, this QS network model comprised 103 biochemical reactions, including DNA transcription, mRNA translation, protein complex formation, inhibition reactions, and molecule-to-molecule interactions with positive, negative, or unknown effects (Figure 2). All reactions and interactions between them had the kinetic rate constant (*k*) values reported in the literature. The values of *k* not reported in the literature were assumed to be within the range of published *k* values for similar reactions (data available in Mendeley Data, https://doi.org/10.17632/2xzzkmnpfx.1). For model construction, a system of 114 ODE was generated using a total of 103 kinetic parameters and represented using the SBGN graphical notation of the SBML models.

**Figure 2 metabolites-13-00659-f002:**
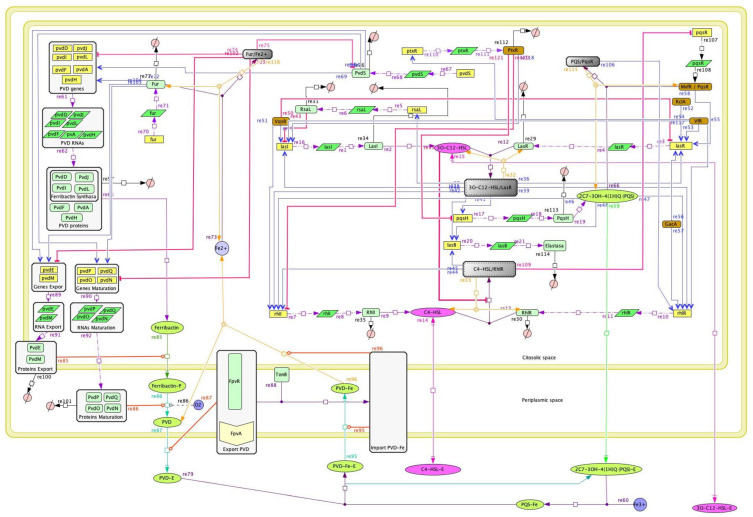
**The quorum-sensing network regulates the expression of pyoverdine in *Pseudomonas aeruginosa*.** The QS network was constructed by analysis of bibliomic information, schematized in CellDesigner 4.4, and modeled with the SBML ODE Solver Library (SOSlib). The chemical species are represented in SBML format: Genes by straight yellow squares; mRNAs by dark green diagonal squares; proteins by light green and brown oval squares; single molecules by bright green and pink ovals; and O_2_, Fe^2+^, and Fe^3+^ ions by blue circles. Gray and white squares represent complexes of simple molecules and macromolecular complexes, respectively. Red circles represent the degradation processes of all proteins included in the model. Arrow colors represent the following interactions between chemical species: Purple for transcription and translation processes; black for complex formation processes; yellow for complex cleavage; pink and green for the diffusion of signal molecules; red and blue for negative and positive regulatory processes, respectively; and orange for protein interactions in specific reactions. (Each symbol and its meaning are available in Mendeley Data, https://doi.org/10.17632/2xzzkmnpfx.1).

The QS network was modeled in the twenty simulation scenarios described in Table 1. The system was set up in all of these scenarios with an initial concentration of 1.0 μM of all QS genes and super-regulator proteins. In addition, extracellular Fe^3+^ availability and PVD transport system proteins were fixed to a starting concentration of 2.0 μM (data available in Mendeley Data, https://doi.org/10.17632/2xzzkmnpfx.1). In all simulation scenarios, the concentration of the E-PQS signal molecule was modified to emulate the behavior of in vitro cultures of *P. aeruginosa*, where population growth increases as a function of time (more cells per unit volume) (Table 1). The performance of the QS network model was understood as synthesis of the chemical species of interest, i.e., intracellular production of the QS signal molecules (homoserine-lactones 3O-C12-HSL and C4-HSL, and quinolone PQS), cytosolic production of ferribactin, and periplasmic production of PVD (Figure 3). 

The first simulation was run under the “initial conditions” scenario (Table 1). This simulation tested the intracellular production of the chemical species of interest (data available in Mendeley Data, https://doi.org/10.17632/2xzzkmnpfx.1). In this simulation, increasing concentrations of QS signal molecules were not considered because the purpose was to assess the basal capacity of the model to reproduce the QS circuits and produce QS signal molecules intracellularly. In silico intracellular production of the chemical species of interest corresponded to simulated concentrations of 0.0234 μM for 3O-C12-HSL, 0.0102 μM for C4-HSL, 0.00099 μM for PQS; 0.0158 μM for ferribactin, and 0.0635 μM for PVD (Figure 3). These values are relevant because they test whether the model behaves in agreement with data reported in the literature [56,57,58].

The low concentration of these chemical species showed that the system on its own produced a basal amount of signal molecules, ferribactin, and PVD in response to the concentration of Fe^3+^ available in the extracellular medium (simulated at an initial concentration of 2.0 μM). Subsequently, simulations run according to the scenarios in Table 1 showed that the higher the concentration of E-PQS, the higher the intracellular PVD production. The experimentation performed allowed the evaluation of the sensitivity of the output variable [intracellular PVD] to marginal changes in the input parameter [E-PQS] (Appendix B).

The highest PVD yield (0.122 μM) was achieved with 0.6 μM E-PQS (Sc16; ID = PQSE06 in Table 1). On the other hand, the highest concentrations of E-PQS (Sc17 to Sc20) were related to a decrease in PVD production (Figure 4A), indicating that the saturation point of the system was reached at 0.6 μM E-PQS. At the saturation point in the cell, in the particular case of the signaling molecule PQS, when there is a PQS “maximum” concentration inside the cell—*which depends directly on the QS phenomenon* [7,8]—no new extracellular PQS molecules can enter the cell, thus decreasing the virulence factors such as QS-regulated PVD production over time [13,14,15,16].

A dynamic equilibrium was reached in each simulation scenario, meaning that all chemical species in the model reached a constant value after some time interval. Contrary to the effect of the initial E-PQS concentration on intracellular PVD production, the final extracellular PVD concentration was not affected in any of the simulated scenarios (Figure 4B). However, the system evidenced a change in the trajectory of extracellular PVD production.

### 3.2. Pseudomonas aeruginosa Metabolic Network Model CCBM1146: An Improved Version of the Genome-Scale Metabolic Model iMO1056

The iMO1056 model, including its biomass reaction [37], was the starting point. First, the data associated with all reactions in the model were reviewed, followed by a reverse-engineering curation process. Next, 90 genes, 120 metabolites, and 41 reactions, including those in the metabolic pathway of PVD synthesis, were selected to fill the gaps in the model. Finally, PVD was added to the biomass reaction as a metabolite to obtain the improved model CCBM1146 (Table 4).

The curation process was complemented by (i) evaluation of metabolite production and consumption in the biomass reaction by reverse engineering; (ii) detection and resolution of pathologies in the 880 metabolites of the network: 29.51% had pathologies, 17.25% were resolved and 12.26% with unresolved pathologies remained in the model; (iii) detection and resolution of TICs: 23 TICs were identified and entirely resolved in the network. Finally, the proposed model, CCBM1146, comprised 1123 reactions, 880 metabolites, 136 protein complexes and transport proteins, and 1146 genes coding for 826 enzymes. Each reaction was manually assigned a metabolic system and its corresponding labeled subsystems according to the ontology pathway (https://biocyc.org/searchhelp.shtml#ontology_search, accessed on 15 October 2018) established by the BioCyc database (Figure 5). Analysis of the BioCyc metabolic system labels revealed that the CCBM1146 model consisted mainly of reactions of central metabolism, such as biosynthesis of cofactors, prosthetic groups, and electron transporters, followed by transport and exchange reactions involved in amino acid biosynthesis and degradation, lipopolysaccharide biosynthesis, and reactions involved in the biosynthesis of lipid molecules for cell wall formation (data available in Mendeley Data, https://doi.org/10.17632/y9htx3fcjm.1).

The definitive CCBM1146 model and the steady-state FBA approximation were used to obtain the flux distribution of reactions in the *P. aeruginosa* metabolic network. The flux value (the rate at which a metabolite is formed as a function of the available biomass [mmol/gWD*h^−1^]) obtained for the objective function of the model was 0.55 h^−1^ (data available in Mendeley Data, https://doi.org/10.17632/y9htx3fcjm.1). After simulation, 352 reactions with active fluxes were obtained, including those for synthesis, transport, and exchange of the QS signal molecules (PQS, 3O-C12-HSL, and C4-HSL) and PVD. This relatively small number of reactions is principally due to the structure of the biomass equation. Also, it could be related to the addition of PVD to the biomass reaction, which triggers the activation of a small set of essential reactions of the central metabolism (including biomass metabolites and PVD synthesis reactions). Furthermore, since PVD is a secondary metabolite, these reactions result in PVD production upon activating necessary intermediary metabolism and secondary metabolism reactions in the model.

### 3.3. Integrative Model Simulations Evidenced the Influence of Quorum-Sensing Signaling on Pyoverdine Biosynthesis in Pseudomonas aeruginosa Cultures

As described in materials and methods, the design of the integrative model required the combination of the QS regulatory gene network model and the *P. aeruginosa* metabolic network model. Since these networks have different types of biological language, the flux values of nine reactions shared by both networks were considered to be useful for merging the behavior of chemical species in both types of network models. These nine shared reactions were involved in the synthesis and transport of QS signal molecules, ferribactin synthesis, and PVD synthesis and transport. After applying the multi-stage FBA approximation in six simulation scenarios of the integrated model (Table 3), the Sc3 scenario was chosen for subsequent simulations. This scenario evidenced the highest sensitivity of the CCBM1146 model to changes in the fluxes of reactions involved in the metabolic synthesis of the three QS signal molecules and transport of PVD. Subsequently, twenty simulations were run under the multi-stage FBA approximation in scenarios with varying initial E-PQS concentration (Table 1) and fixing the flux values of four of the reactions shared by the QS and the *P. aeruginosa* metabolic network models included in the Sc3 scenario (Table 3). Figure 6A shows a dynamic response of the integrative model objective function in the first four hours of in silico simulations; this response was related to changes in signal intensity derived from the QS model (deterministic model). This response was evident in all simulation conditions of the CCBM1146 model. Therefore, the sensitivity of the CCBM1146 model can be biologically interpreted as follows: the higher the E-PQS concentration at the beginning of bacterial growth, the higher the bacterial metabolic demand, and the greater the deceleration of the objective function (biomass maximization) (Figure 6A).

Twenty simulations run using the DFBA approximation and under conditions identical to those of the multi-stage FBA approximation showed similar results (Figure 6B), supporting the consistency of the proposed CCBM1146 model. During the first hours of simulation, the objective function showed a dynamic response related to changes in the signal intensity derived from the QS network in all simulation scenarios, i.e., as the QS signal increased, the behavior of the objective function changed. Subsequently, the objective function reached the same value in all simulation scenarios at approximately the 14-h interval. Finally, the value of the objective function decreased to zero simultaneously with the overall consumption of the available glucose (carbon source) in the model.

In addition, biomass profiles and concentrations of metabolites of interest were obtained under the DFBA approximation. Both in silico (Figure 7A) and in vitro (Figure 7B) data showed an exponential increase in biomass profiles up to hour 14 of culture of *P. aeruginosa.* A similar exponential increasing trend in biomass concentration was also observed over time up to the stationary phase. Therefore, it is possible to suggest that the biomass profile predicted (Figure 7A) by the CCBM1146 model properly represents the in vitro behavior of *P. aeruginosa* (Figure 7B).

In silico and in vitro PVD profiles were also compared (Figure 8). In silico, the PVD value increased in the first 8 h of simulated culture and reached a stable concentration around 9 h (Figure 8A). The in vitro PVD profile showed an increasing trend in PVD concentration over time. The maximum PVD concentration was reached around 10 h of culture. It coincided roughly with the mid-exponential phase of bacterial growth in vitro (Figure 8B) when signal molecules were produced. Bacteria responded to those signal molecules in the stationary phase [59].

However, a variation in PVD concentration was observed in vitro, probably related to typical PVD oxidation over time in a natural system. Moreover, the oxidation changed the characteristic green color of PVD in vitro due to the quinoline-based cyclic fluorescent chromophore, which is responsible for the bright fluorescence of pyoverdine [60].

In silico, glucose concentration varied over time in all simulation conditions (Figure 9A); the initial glucose concentration (55 mM) was consumed entirely at approximately 14 h. In addition, in silico profiles of the QS signal molecules, i.e., 3O-C12-HSL (Figure 9B), C4-HSL (Figure 9C), and PQS (Figure 9D), also showed their concentrations increased up to hour 10. This time roughly corresponded to the mid-exponential phase of *P. aeruginosa* growth in vitro and agreed with data from the literature, according to which cells have a basal production of signal molecules that reach a maximum concentration in the mid-exponential phase of growth.

## 4. Discussion

A thorough understanding of complex biological systems’ behavior requires studying and comprehending the interplay of several processes that often occur at different spatial and temporal scales in the biological system as a whole [1,2,61]. The biological problem addressed in this work was the synthesis of the virulence factor PVD in *P. aeruginosa*. The synthesis of PVD requires bacterial communication through the QS phenomenon to produce signal molecules that regulate the expression of *pvd* genes encoding enzymes that catalyze the synthesis of the virulence factor PVD. Three types of biological processes are involved in PVD synthesis, namely: (i) the QS signaling pathway, (ii) a regulatory pathway for *pvd* gene expression, and (iii) the effect of these two processes on the *P. aeruginosa* metabolic network for PVD biosynthesis. These three processes were encompassed here in a multiscale model to explain the influence of the QS phenomenon on the metabolic pathway of PVD biosynthesis in *P. aeruginosa*.

### 4.1. The QS Gene Regulatory Network Model Emulates the Natural Behavior of Pseudomonas aeruginosa

The deterministic in silico model of the QS gene regulatory network in *P. aeruginosa* proposed in this work is the first approximation that includes the three QS systems reported for this microorganism. It also incorporates the autoregulatory mechanisms of these three QS systems as well as the mechanisms responsible for the expression of *pvd* genes, the transport of PVD to the extracellular space, and the chelation and entry of iron into the bacterium through the PVD/Fe^3+^ complex. The data obtained from QS model simulations are supported by the fact that any system in dynamic equilibrium tends to reach a steady state, which translates into an equilibrium that resists external forces of change (perturbations). When the system is perturbed, the regulatory systems—in this case, the QS systems—respond to the signals emitted to establish a new equilibrium; this function is executed by the QS systems themselves and is known as feedback control. All biological processes that integrate and coordinate the functioning of living organisms are examples of the homeostatic regulation required by the system to ensure cell survival. Thus, the biological system is driven to the homeostatic state expected from its behavior, which is likely to maintain stability while adapting to the optimal conditions for survival. This argument is supported by the work of Gonzalez et al. [62], which states that the steady state is the inherent state of biological systems in the environment. Therefore, analyses of bacterial systems during the steady state have been the basis for most conclusions on bacterial network modeling [62].

The proposed deterministic QS network model emulated in silico the natural behavior of *P. aeruginosa* in culture, i.e., as the bacterial population increased (more cells per unit volume), the extracellular concentration of QS signal molecules per unit volume also increased. It was also evident that the signal molecules of all three QS systems were synthesized intracellularly and that as their concentration decreased, PVD production increased; that is, the QS network model reproduced known biological behavior and worked according to the expected cellular dynamics concerning the QS systems (Figure 3). It is important to note that a complex modeling of the internal regulation of the QS network in *P. aeruginosa* was achieved, as well as the possibility to simulate extracellular conditions that can lead to an increase in bacterial population density by modifying the concentration of the exogenous QS signal molecules. As shown in Figure 4A, the simulated intracellular PVD production was directly proportional to the extracellular PQS concentration (increase in QS strength) under the simulation conditions posed for each scenario (Table 1). However, after reaching a peak concentration, PVD production decreased, indicating that the system reached a saturation point with the following range of dynamic equilibrium. This result could be related to an increase in the intracellular Fe^2+^ concentration resulting from the autoregulatory mechanisms of the system. In the cytosol, the ferrous ion (Fe^2+^) forms a complex with the Fur protein, Fur/Fe^2+^, that binds to the promoter regions of *pvd* genes (iron-repressible or iron-regulated genes), thus inhibiting their transcription. Under limiting intracellular iron concentrations, the Fur-mediated repression decreases, and a positive transcriptional regulation of *pvd* genes ensues [63,64,65]. The proposed deterministic QS network model reproduces this regulatory phenomenon that is directly related to the production of PVD, the formation of the PVD/Fe^3+^ complex at the extracellular level, its subsequent entry into the cell via an ABC-type transport system, and finally, to the release of Fe^2+^ into the cytosolic space.

On the other hand, it should be noted that the simulated extracellular PVD (Figure 4B) behaved differently from intracellular PVD in response to the increase of extracellular PQS levels. In the first few hours, the system showed a change in the trajectory of the extracellular PVD concentration. This different dynamic response could be attributed to the PVD transport mechanism involving the surface receptor FpvA, the anti-sigma factor FpvR [66,67], and the protein responsible for energy transduction to import molecules through the TonB protein located in the outer membrane of the bacterium [7,8,14,15,67,68,69]. According to Arevalo-Ferro et al., the expression of TonB is also regulated by QS in *P. aeruginosa* [70]. Thus, the results of simulations performed with the QS Network can be supported, as mentioned previously, on the basis that any system in dynamic equilibrium tends to reach a stable state, which translates into an equilibrium that resists external forces of change. 

In addition, the model acquired the emerging property of resilience, characteristic of adaptive biological systems. Although this property was not modeled, it became apparent when the model was disturbed as the QS regulatory system responded to outputs to establish a new dynamic equilibrium that reached a steady state. The generated equilibrium resisted disturbances, understood as external forces of change. The in silico results were similar to those reported on the natural behavior of QS mechanisms in *P. aeruginosa* [67,71].

### 4.2. The Proposed Integrative CCBM1146 Model Helps to Infer the Influence of the QS Phenomenon on the PVD Metabolic Biosynthesis in Pseudomonas aeruginosa

This study proposed a systemic approach to design an integrative multiscale model to understand the influence of the QS phenomenon on the expression of the PVD metabolic phenotype of *P. aeruginosa* by combining two networks with different temporal and spatial scales, namely, the QS communication network responsible for regulating the expression of *pvd* genes and the bacterial metabolic network of PVD biosynthesis.

The CCBM1146 metabolic network model is an improved version of the genome-scale metabolic network iMO1056 developed by Oberhardt et al. [37]. The CCBM1146 model includes reactions of the central metabolism, all reactions for the biosynthesis of PVD and QS signal molecules, as well as those involved in their corresponding transport and exchange. The strategy used to combine the two networks was based on obtaining from the QS network simulations the fluxes of the reactions shared by the two networks to set them in the corresponding metabolic reactions in the CCBM1146 model as constraints of the optimization problem in the DFBA. The network combination strategy was evaluated by simulations in the integrative CCBM1146 model under a multi-stage optimization approach to obtain the values of changes in the objective function (biomass maximization) over time without considering the changes in metabolite concentrations. Subsequently, the model was run under a DFBA approximation to obtain the distribution of reaction fluxes over time and the concentration profiles of biomass and metabolites of interest. As shown in Figure 6A, the results of the multi-stage FBA modeling evidenced a different dynamic response of the objective function related to changes in the signal strength from the QS network model in all simulation scenarios. As the QS signal increased, there was a noticeable change in behavior of the objective function, which decreased to a minimum and then increased to a stable value in all simulation scenarios. The decrease in the objective function value (Figure 6A) could be attributed to its degeneracy by one of the elements involved in the optimization, which does not support its value to increase. Degeneracy also occurs when the constraint in the optimization problem increases [72,73]. According to the study by Wintermute et al., “the FBA generally cannot predict a unique rate for all fluxes. A solution that maximizes growth rate is typically mathematically degenerate, describing a region in flux space rather than a single point. Solution degeneracy is a well-described problem in systems like metabolism, which are flexible, internally redundant, and underdetermined by data” [72]. Alternatively, it may also occur because the availability of resources does not imply that the model immediately exploits them in favor of maximizing the objective function. Instead, it can be interpreted as a period of adjustment or adaptation of all variables to optimize the maximization of the objective function. However, as the influence of the constraint increases—understood as a more significant influence of the QS signal—the point at which the value of the objective function does not vary significantly over time (steady state) is reached in a shorter time.

The DFBA results were similar to those from the multi-stage FBA approximation (Figure 6B). However, under the DFBA approximation, the objective function reached a stable value around the hour 14 and then dropped to zero in all simulation scenarios. The point at which the objective function reached a zero value coincided with the point of depletion of the carbon source (glucose). This fact could explain the results, considering that once the total glucose consumption is reached, the maximization of the objective function cannot take a value other than zero. In in vitro cultures, bacterial growth reaches the stationary phase when the nutrients in the medium—especially the carbon source—have been consumed, indicating that the bacteria can no longer grow. Growth curves of *Pseudomonas aeruginosa* in cultures incubated under controlled conditions—intended to validate some of the results of the CCBM1146 model—showed that the stationary phase was reached between 14 h and 15 h of culture (Figure 10), similar to the in silico point at which the value of the objective function dropped to zero.

The divergences observed between the in vitro and in silico results may be due to several factors, including the growth conditions of each system. For the in silico model, glucose was considered the sole carbon source; its concentration, as well as those for nitrogen, phosphate, and sulfate sources, were estimated from the composition of the LB culture medium. The LB culture medium is a complex mixture of nutrients because its main constituents are yeast extract and tryptone, so the concentration of each nutrient could not be accurately determined. In contrast, in the in vitro model, the bacteria grew in cultures prepared with LB medium. There, they could take advantage of the total concentration of nutrients and utilize carbon sources other than glucose, which may be reflected in bacterial growth. Furthermore, it is important to note that the in vitro growth conditions were optimal for the microorganism, even better than those in vivo. 

Finally, it should be noted that in the in vitro cultures, the bacterial population produced the QS signal by itself; i.e., the culture medium was not supplemented with synthetic QS signal molecules. Nevertheless, the pattern of behavior of the biomass profile (Figure 7B), and PVD concentration profile (Figure 8B) were adequately reproduced by the in silico results obtained in the integrative model under the DFBA approximation (Figure 7A and Figure 8A).

The methodological strategy employed in this study and the results indicate that the proposed multiscale CCBM1146 model offers valuable insights into the influence of the QS phenomenon on the metabolism and subsequent behavior of *P. aeruginosa* and other microorganisms related to the synthesis of virulence factors such as PVD. In addition, the CCBM1146 model could help to infer the metabolic behavior of *P. aeruginosa* that arise in the presence of different concentrations of QS signal molecules. Therefore, the present study also can be a starting point for comprehending the biosynthesis of other siderophores regulated by QS in *P. aeruginosa*, such as Pyochelin (PCH), which, like PVD, has a high iron affinity, and the newly discovered narrow-spectrum metallophore Pseudopaline (PSP), that acts as a virulent factor too and is involved in nickel and zinc uptake [6,75,76]. This study made possible the design of an original methodology based on tools from different areas of knowledge, such as biological network reconstruction methods from systems biology, genomics, and bibliomics, as well as engineering methods, such as optimization, from the area of logistics and operation research; together, they provided a way to study the hierarchies of biological systems in a unified way, according to the principles of systems biology.

## 5. Conclusions

From a systemic perspective, this work presents a methodological strategy to design a novel multi-scale and multi-class model proposal by combining two classes of models with different scales. The CCBM1146 model helped to characterize and explain the complex and emerging behavior derived from the interactions between these two models, which would have been impossible by studying each model or scale separately. Moreover, the CCBM1146 model served to understand and infer the natural influence of PQS intensity on the metabolic PVD biosynthesis in *P. aeruginosa*. In the future, this integrative model could help to infer the different metabolic phenotypes that may arise among other microorganisms when exposed to different concentrations of the signal molecules of their own QS circuits responsible for regulating the synthesis of their virulence factors. Furthermore, the DFBA approximation applied to the CCBM1146 model evidenced its descriptive capability for the profiles of biomass and PVD, showing a similar behavioral pattern of an in vitro culture of *P. aeruginosa*; this makes it more valuable and suitable for evaluating, *in the future and from holistic and dynamic perspectives*, how different conditions can affect the behavior of *P. aeruginosa* for the synthesis of different virulence factors regulated by QS. 

Moreover, the model could be used to test experimental conditions in vitro. Finally, this work proposes, for the first time, the integration of the quorum-sensing gene regulatory network with the *P. aeruginosa* metabolic network for PVD biosynthesis. Combining these two network models was possible by fixing the fluxes of reactions shared by both models as system constraints in the multi-stage FBA and DFBA approximation. Thus, it was possible to model the influence of the QS phenomenon on the *P. aeruginosa* metabolism to PVD biosynthesis as a function of QS signal intensity.

However, though in this work a sensitivity analysis was performed for the QS network model, for further modeling works under the steady-state and dynamic-state flux balance analysis approach of the CCBM1146 integrated model, it is recommended that a sensitivity analysis be performed.

## Figures and Tables

**Figure 1 metabolites-13-00659-f001:**
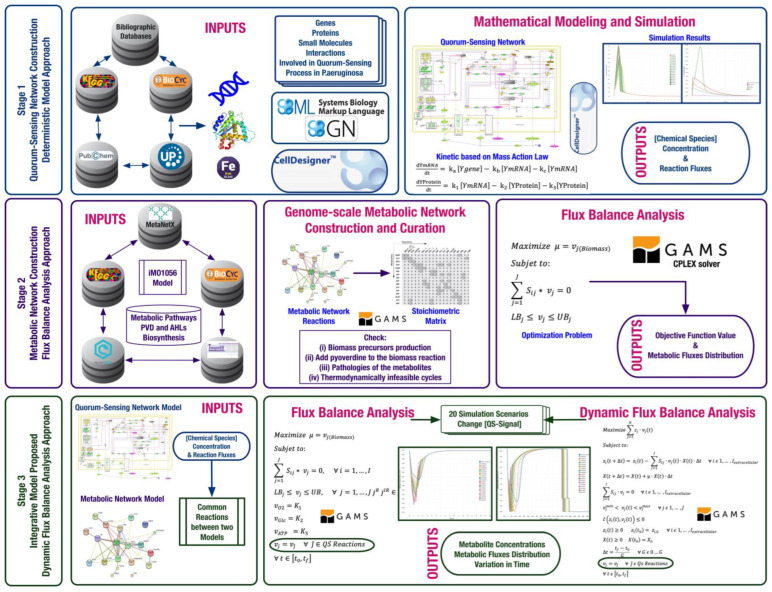
**Methodological workflow for designing a computational model to infer the influence of the quorum-sensing phenomenon on the PVD metabolic synthesis in *Pseudomonas aeruginosa* from a systemic perspective. Stage 1**. Construction of the QS gene regulatory network under a deterministic approach using a kinetics based on the mass action law. **Stage 2**. Construction of the *P. aeruginosa* metabolic network using the flux balance analysis approach. **Stage 3**. Integration of the QS network and the *P. aeruginosa* metabolic network into a multiscale model to be proposed and modeled under a dynamic flux balance analysis approach.

**Figure 3 metabolites-13-00659-f003:**
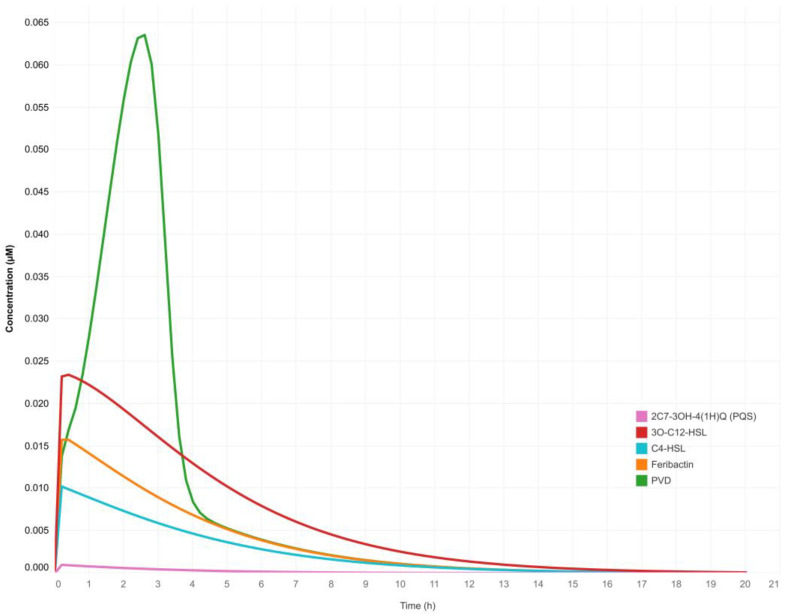
**Simulation of the quorum sensing gene regulatory network model of *Pseudomonas aeruginosa*.** Intracellular production of QS signal molecules (3O-C12-HSL, C4-HSL, PQS), ferribactin, and PVD under the “initial conditions” scenario (Table 1).

**Figure 4 metabolites-13-00659-f004:**
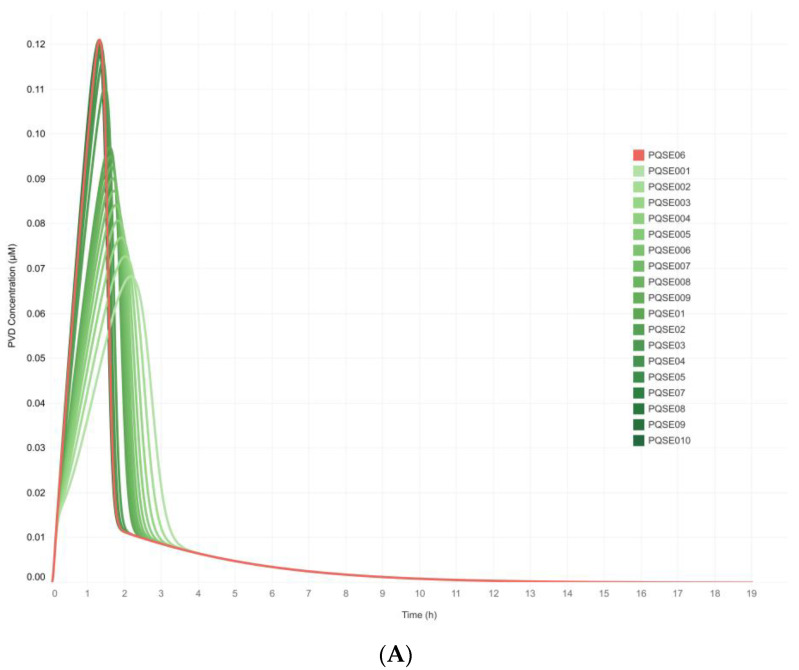
**Simulated** (**A**) **intracellular and** (**B**) **extracellular pyoverdine production in Pseudomonas aeruginosa cultures varied as a function of initial E-PQS concentration**. Simulations were run in 20 scenarios at varying initial concentrations of extracellular PQS (from 0.01 μM to 0.1 μM with 0.01-unit intervals and from 0.1 μM to 1.0 μM with 0.1-unit intervals (Table 1).

**Figure 5 metabolites-13-00659-f005:**
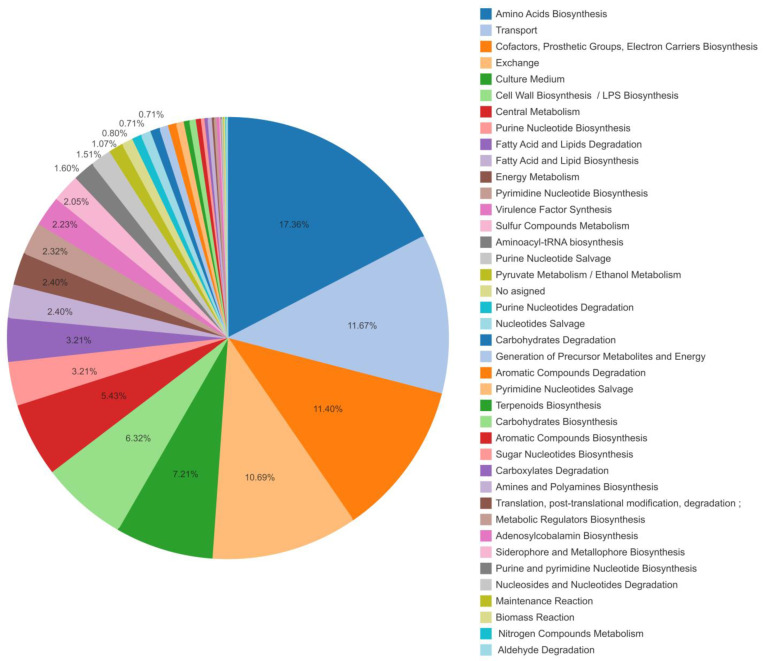
***Pseudomonas aeruginosa* metabolic network model CCBM1146 for pyoverdine biosynthesis involves reactions of the central metabolism and secondary metabolism**. The most representative metabolic systems corresponded to the biosynthesis of cofactors, prosthetic groups, and electron transporters, followed by reactions responsible for metabolite transport and exchange, and then amino acid biosynthesis and degradation. Another group of reactions was involved in the synthesis of lipopolysaccharides and other lipid molecules for cell wall formation. The reactions in the model were classified according to the metabolic system to which they are ascribed in the BioCyc database (data available in Mendeley Data, https://doi.org/10.17632/y9htx3fcjm.1).

**Figure 6 metabolites-13-00659-f006:**
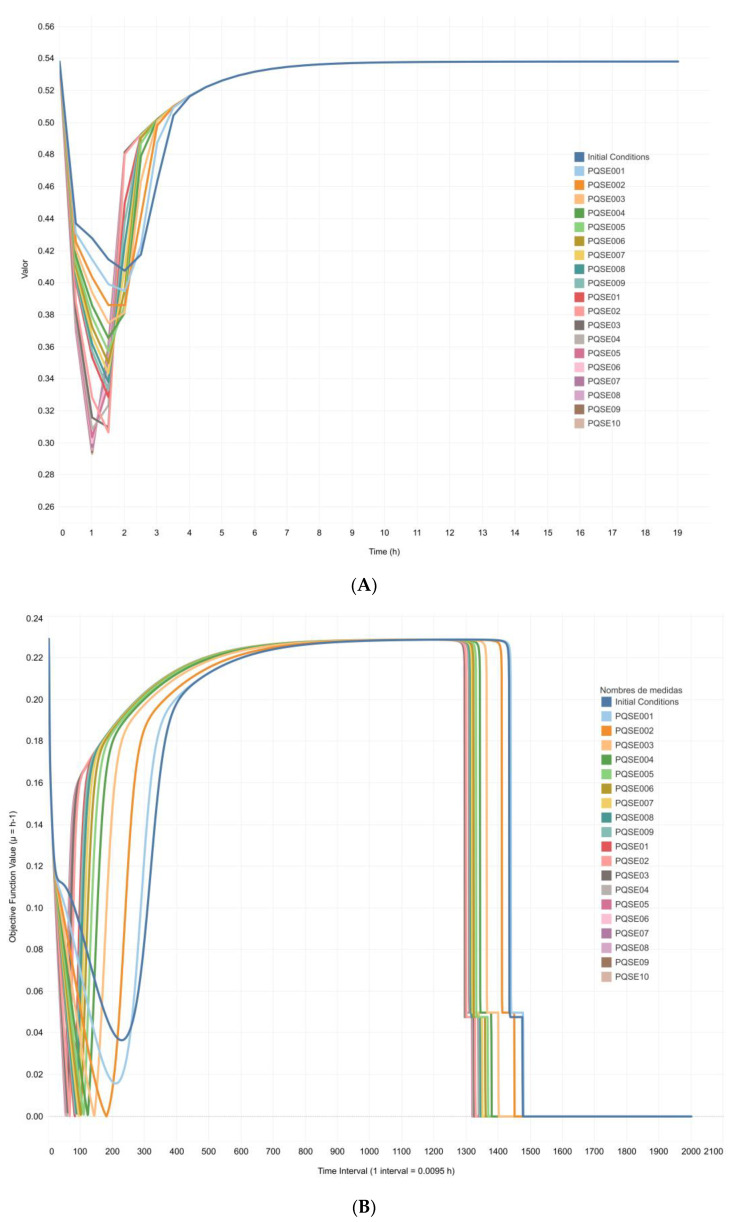
**In silico profile of the CCBM1146 model objective function under** (**A**) **the multi-stage FBA approximation and** (**B**) **the dynamic FBA approximation**. Simulation results under the multi-stage FBA approximation were generated in GAMS for the CCBM1146 model at varying initial E-PQS concentrations (Table 1) and fixing the flux values of four reactions shared by the QS and *P. aeruginosa* metabolic network models in the Sc3 scenario (Table 3).

**Figure 7 metabolites-13-00659-f007:**
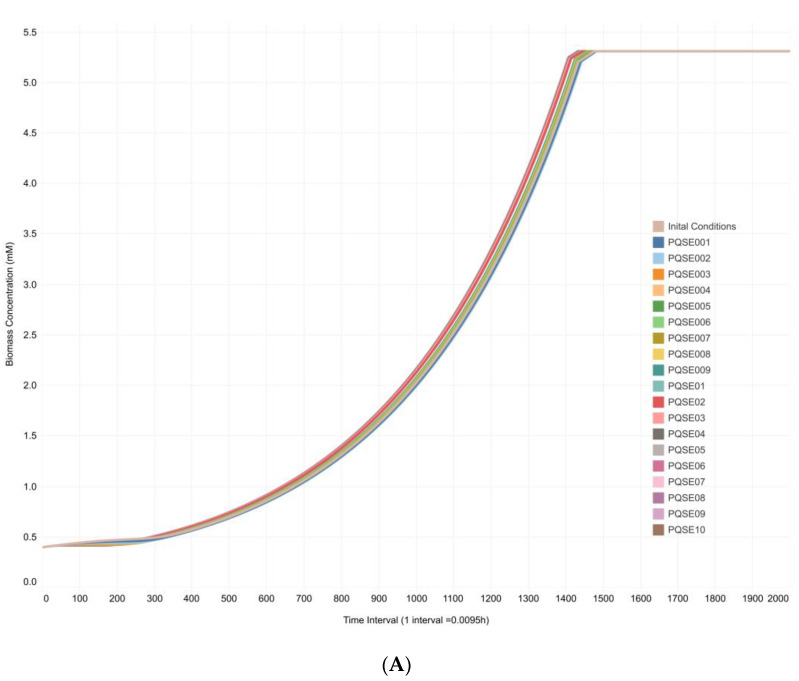
**In silico and In vitro biomass profiles of *Pseudomonas aeruginosa* cultures.** (**A**). In silico biomass profile of *P. aeruginosa* cultures in the CCBM1146 model under the DFBA approximation. Simulation results were generated in GAMS for the CCBM1146 model at varying initial E-PQS concentrations (Table 1) and fixed flux values of four reactions shared by the QS network and *P. aeruginosa* metabolic network models in the Sc3 scenario (Table 3). (**B**). In vitro biomass profile of *P. aeruginosa* cultures. This biomass profile was generated from average dry weight data of *P. aeruginosa* cultures grown under controlled laboratory conditions.

**Figure 8 metabolites-13-00659-f008:**
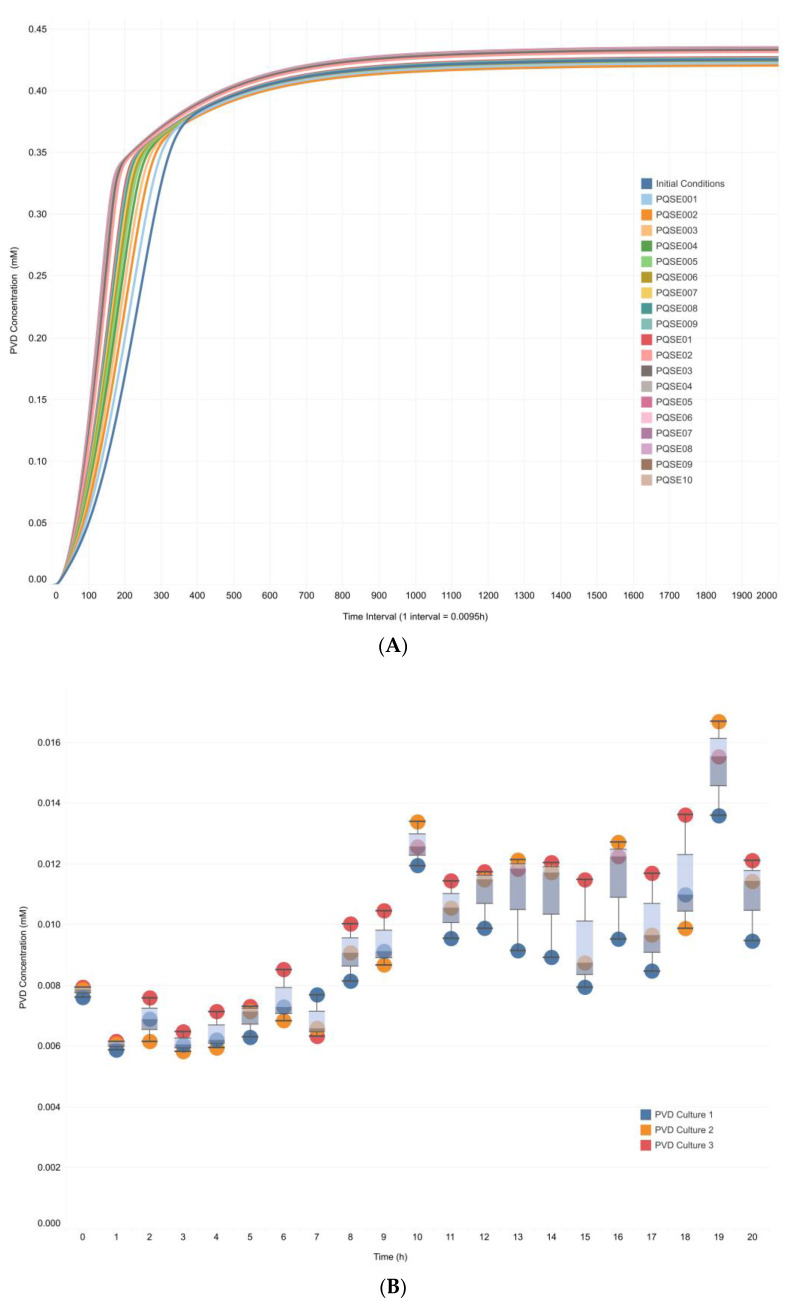
**In silico and In vitro profiles of pyoverdine production in *Pseudomonas aeruginosa* cultures.** (**A**). In silico profile of pyoverdine production in the CCBM1146 model under the DFBA approximation. Simulation results were generated in GAMS for the CCBM1146 model at varying initial E-PQS concentrations (Table 1) and fixed flux values of four reactions shared by the QS network and *P. aeruginosa* metabolic network models in the Sc3 scenario (Table 3). (**B**)**.** In vitro profile of pyoverdine production. This profile was generated from the average absorbance (450 nm) data [53,55] of cultures of *P. aeruginosa* grown under controlled laboratory conditions.

**Figure 9 metabolites-13-00659-f009:**
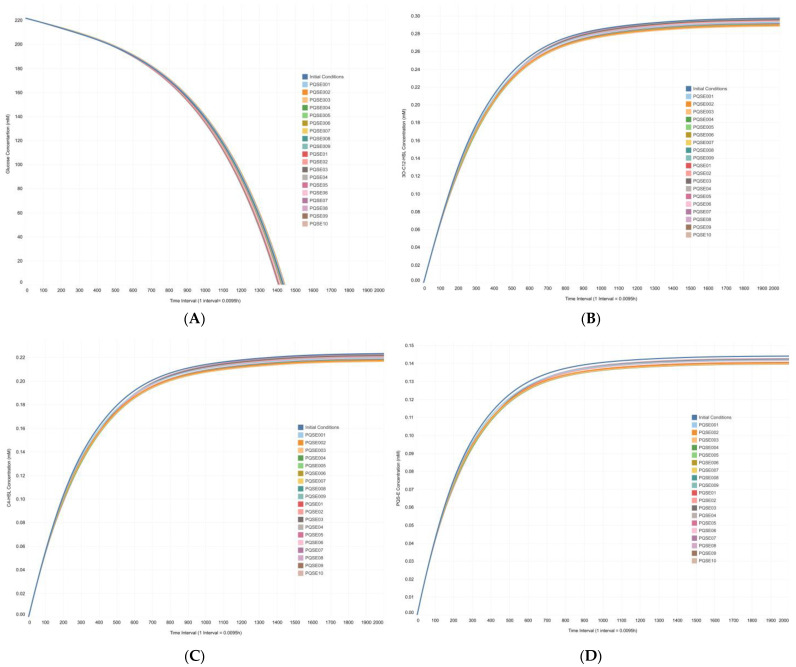
**In silico** (**A**) **glucose profile,** (**B**) **QS signal molecule 3O-C12-HSL profile,** (**C**) **QS signal molecule C4-HSL profile, and** (**D**) **extracellular QS signal molecule PQS in *Pseudomonas aeruginosa* under the DFBA approximation.** Simulation results were generated in GAMS for the CCBM1146 model at varying initial E-PQS concentrations (Table 1) and fixed flux values of four reactions shared by the QS network and *P. aeruginosa* metabolic network models in the Sc3 scenario (Table 3).

**Figure 10 metabolites-13-00659-f010:**
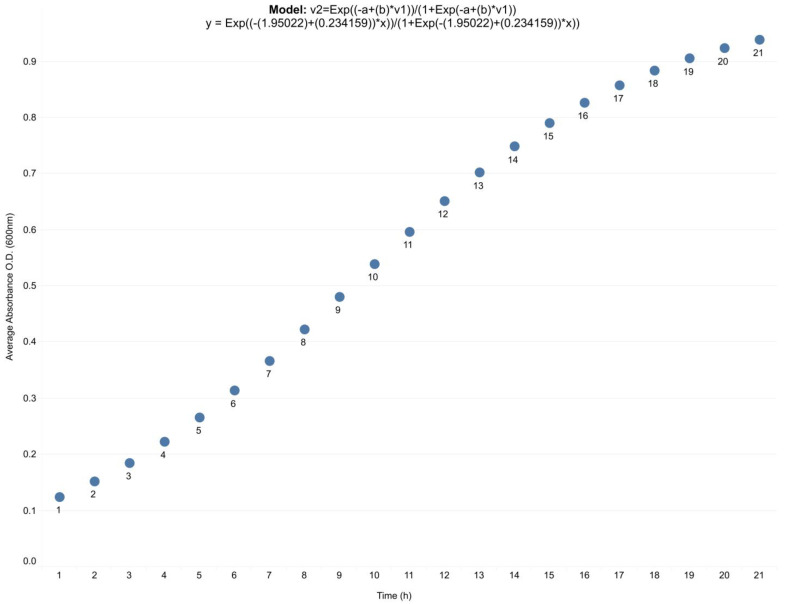
**In vitro growth profile of *Pseudomonas aeruginosa*.** The growth curve shows an exponential phase starting at around hour 6 and a stationary phase between 19 h and 20 h, in agreement with Kim et al. [74]. The growth curve was generated from mean OD (600 nm) data of *P. aeruginosa* cultures (*n* = 3) that were analyzed by logistic regression. In the model equation, v2 represents the fitted absorbance (y-axis, dependent variable), and v1 (x-axis) represents time.

**Table 1 metabolites-13-00659-t001:** Simulation scenario conditions for the quorum-sensing network model.

Simulation Scenario	E-PQS [μM]	ID Simulation Scenario	Simulation Scenario	E-PQS [μM]	ID Simulation Scenario
Sc1	0.00	Initial conditions	Sc11	0.1	PQSE01
Sc2	0.01	PQSE001	Sc12	0.2	PQSE02
Sc3	0.02	PQSE002	Sc13	0.3	PQSE03
Sc4	0.03	PQSE003	Sc14	0.4	PQSE04
Sc5	0.04	PQSE004	Sc15	0.5	PQSE05
Sc6	0.05	PQSE005	Sc16	0.6	PQSE06
Sc7	0.06	PQSE006	Sc17	0.7	PQSE07
Sc8	0.07	PQSE007	Sc18	0.8	PQSE08
Sc9	0.08	PQSE008	Sc19	0.9	PQSE09
Sc10	0.09	PQSE009	Sc20	1.0	PQSE10

**Table 2 metabolites-13-00659-t002:** Reactions shared by the Quorum-Sensing and *Pseudomonas aeruginosa* metabolic network models.

Quorum-Sensing Network	Metabolic Network	EC/TC Number	Reaction Equation in the Metabolic Network Model
3O-C12-HSL production	3O-C12-HSL synthesis	EC 2.3.1.184	[c]: 3oxddACP + amet <==> 5mta + ACP + h + n3oxdd-hsl
C4-HSL production	C4-HSL synthesis	EC 2.3.1.184	[c]: amet + butACP <==> 5mta + ACP + h + nb-hsl
PQS production	PQS synthesis	EC 1.14.13.182	[c]: fadh2 + h + hhq + o2 --> nad + h2o + pqs
C4-HSL diffusion	C4-HSL transport	-	nb-hsl[c] <==> nb-hsl[e]
3O-C12-HSL diffusion	3O-C12-HSL transport	-	n3oxdd-hsl[c] <==> n3oxdd-hsl[e]
PQS diffusion	PQS transport	-	pqs[c] <==> pqs[e]
Ferribactin production	Ferribactin synthesis	EC 6.3.2.	[c]: glu-L + tyr-L + (2) ser-L + arg-L + 24dab + (2) fohorn + lys-L + (2) thr-L --> fbn + (12) h2o + (2) h
PVD production	PVD synthesis	EC 1.14.18.	[c]: fbn + o2 --> pvd1 + h2o
PVD export	PVD transport	TC-1.B.14.1.6	pvd1[c] --> pvd1[e]

**Table 3 metabolites-13-00659-t003:** Biochemical reactions involved in the integrative model simulation scenarios using FBA and DFBA approximations. Sc: Scenario number; PVD: pyoverdine.

Metabolic Reaction	Sc1	Sc2	Sc3	Sc4	Sc5	Sc6
3O-C12-HSL synthesis	X	X	X			
C4-HSL synthesis	X	X	X			
PQS synthesis	X	X	X			
C4-HSL transport				X	X	X
3O-C12-HSL transport				X	X	X
PQS transport				X	X	X
Ferribactin synthesis	X			X		
PVD synthesis		X			X	
PVD transport			X			X

**Table 4 metabolites-13-00659-t004:** Comparison of *Pseudomonas aeruginosa* metabolic network models iMO1056 and CCBM1146 (data available in Mendeley Data, https://doi.org/10.17632/y9htx3fcjm.1).

Reactions and Components	Model	
	iMO1056	CCBM1146
Metabolic reactions	728	774
Transport reactions	150	146
Biomass reaction	1	1
Maintenance reaction	1	1
Exchange reactions	118	120
Reactions for metabolite input from the culture medium	84	81
Total reactions	1082	1123
Total metabolites	760	880
Total genes	1056	1146

## Data Availability

Quorum-Sensing Model for the Pyoverdine Expression in *P. aeruginosa.*
https://doi.org/10.17632/2xzzkmnpfx.1 (https://data.mendeley.com/datasets/2xzzkmnpfx), and *P. aeruginosa* Genome-scale Metabolic Network-CCBM1146. https://doi.org/10.17632/y9htx3fcjm.1 (https://data.mendeley.com/datasets/y9htx3fcjm). Details data can be available in the Appendix A.

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
