# Peer review of "A Holistic Approach from Systems Biology Reveals the Direct Influence of the Quorum-Sensing Phenomenon on Pseudomonas aeruginosa Metabolism to Pyoverdine Biosynthesis"

_metabolites, 2023, doi:10.3390/metabo13050659_

Round 1

Reviewer 1 Report

A Holistic Approach from Systems Biology Reveals the Direct Influence of the Quorum-Sensing Phenomenon on Pseudomonas aeruginosa Metabolism to Pyoverdine Biosynthesis.

Diana Carolina Clavijo-Buriticá, Catalina Arévalo-Ferro, and Andrés Fernando González Barrios

This work aimed simulate the virulence factor pyoverdine (PVD) biosynthesis in Pseudomonas aeruginosa in non-steady state. For this objective two biological network models were used. The simulation results obtained with a QS gene regulatory network model were used as constraints for the flux balance analysis (FBA) of a genome scale model. The specific growth rate of the biomass as objective function to obtain reaction fluxes through the metabolic network.

The authors need to show that the results obtained with QS gene regulatory network model are valid. Also results obtained in DFBA seem not considered the reduction of specific growth rate due to nutrient consumption.

There are some other aspects in the manuscript that need clarification before it can be considered for publication.

Other comments are:

Line 173-175: Which was the size of the system of ODE? How many parameters were needed to solve this system?

Line 198-200: Please explain why 20 initial values were needed? Relate these values with reported experimental data.

Line 201: Please explain why the rate of these cell reactions occurs at non steady state while all the reactions related with cell growth are at a steady state?

Line 208: Please explain which are the differential equations representing the metabolic pathways that need to be solved?

Line 271 and 293: Please explain the meaning of The fluxes for the QS reactions obtained in the deterministic model in CellDesigner of the QS network (???? ?−1?−1) were adjusted to (???? ???−1â„Ž−1). These are different magnitudes.

Line 276-277: A reference is needed; what is the known cellular behavior?

Line 278: Please explain why these are different scenarios? What is the objective of the simulations?

Line 350: The simulation extended from 0 h to 19 h (2000 intervals, each 0.0095 h). This was already mentioned in line 336.

Line 337: Replace bacterial growth by specific growth rate, and biomass production by biomass concentration.

Line 349: How sensible are the simulation results of the volumetric mass transfer coefficient? Why this value was selected?

Line 354: Why the cell growth experiment was in vitro?

Line 369: A parenthesis is missing in the equation.

Line 391: Please show the total number of parameters and the percentage of these that are known. Please explain why the equations in QS network cannot be added to the GEM.

Line 421-423: Please explain the meaning of the phrase in these lines.

Line 433: A reference is needed.

Line 457: How do simulation results compare with experimental data?

Line 496: It is not clear which was the impact of the changes introduced to the previous model, beside the PVD pathway.

Line 505: The number of active reactions were not only related to the PVD production but also the production of biomass components.

Line 507-508: An explanation is needed here.

Line 525: If the simulation in Figure 7 correspond to a batch process (time dependent) then why the specific growth rate decreased and then increased? It should decrease always since nutrients are being consumed. Related to the effect of E-PSQ a similar trend was also obtained with initial conditions that correspond to?

Line 533: Were the results obtained considering constant consumption rates of the carbon and nitrogen sources?

How do the authors modeled the decrease of the nutrient consumption rates as a function of the concentration of nutrients that decreased.

How were the initial conditions (concentrations) determined? It is not clear what type of process is being simulated.

539-541: The fact that similar results were obtained in the 20 simulations does not probe the consistency of the model? How do the results compare with experimental data?

Line 562: How was the concentration of the biomass (simulated and experimental) transformed in mM units?

Line 703: A reference is needed.

Line 728-739: The objective function is unique; this value can be obtained by a different combination of the other fluxes. However, fluxes related to QS were estimated from the other model and these are fixed.

Line 775-778 and 802: It is not clear how DFBA was able to reproduce the experimental results; which was the error?

Author Response

Dear

METABOLITES Reviewer 1

Section: Bioinformatics and Data Analysis

We are very grateful for the review process carried out on the manuscript entitled: A Holistic Approach from Systems Biology Reveals the Direct Influence of the Quorum-Sensing Phenomenon on Pseudomonas aeruginosa Metabolism to Pyoverdine Biosynthesis - METABOLITES- 2326047.  Thank you so much for your comments. Following your suggestions, we addressed the minor and major changes to the document. These changes and responses to each point are described in the attachment.

Please check the responses in the document.

We would like to thank the referee for evaluating our manuscript. We have tried to address all the reviewers’ concerns in a proper way and believe that our manuscript has been significantly improved.

On behalf of my coauthors,

Diana Carolina Clavijo B. PhDc

Corresponding author.

Reviewer 2 Report

In this work Gonzalez-Barrios and coworkers present a multiscale computational model to explain the synthesis of the siderophore pyoverdine (PVD), which involves the PDV gene expression and the metabolic pathway.

This study is very complete and we consider that it deserves to be published in Metabolites with minor revisions.

Page 10:

"... initial source concentrations of carbon (glucose = 55.5mM), nitrogen (NH4 = 104.127mM), phosphate (Pi = 3.88mM), sulfate (SO4 = 0.286mM), biomass (X = 0.1g/L) and oxygen (O2 = 1mM), as well as the kLa reported by Mahadevan et al."

Ammonia and sulfate formulas are wrong.

Equations in Figure 1 are too small. This figure should be reformulated.

Some computational efforts to explain biological systems should be mentioned:

J. Phys. Chem. B 2023, 127, 5, 1110–1119

J. Chem. Theory Comput. 2012, 8, 9, 3314–3321

J. Am. Chem. Soc. 1998, 120, 37, 9401–9409

Some perspectives should be included

Author Response

Dear

METABOLITES Reviewer 2

Section: Bioinformatics and Data Analysis

We are very grateful for the review process carried out on the manuscript entitled: A Holistic Approach from Systems Biology Reveals the Direct Influence of the Quorum-Sensing Phenomenon on Pseudomonas aeruginosa Metabolism to Pyoverdine Biosynthesis - METABOLITES- 2326047.  Thank you so much for your comments and for considering that our work is complete and deserves to be published in Metabolites Journal. Following your suggestions, we addressed the minor and major changes to the document. These changes and responses to each point are described in the attachment.

Please check the responses in the document.

We would like to thank the referee for evaluating our manuscript. We have tried to address all the reviewers’ concerns in a proper way and believe that our manuscript has been significantly improved.

On behalf of my coauthors,

Diana Carolina Clavijo B. PhDc

Corresponding author.

Reviewer 3 Report

Dear Authors,

The manuscript entitled "A Holistic Approach from Systems Biology Reveals the Direct Influence of the Quorum-Sensing Phenomenon on Pseudomonas aeruginosa Metabolism to Pyoverdine Biosynthesis." has been reviewed.

This article deserves attention since it highlights an important topic related to the importance of in silico studies in different domains of life, from the medical field to the biological one. In this original work authors compare and show by using computational methods the relation between Quorum Sensing (QS) and Pyoverdine PVD biosynthesis in one of the most studied pathogenic bacteria Pseudomonas aeruginosa (P. aeruginosa).

The article is well written in English, well presented, and I think it will be very attractive for readers. But I have some comments (minors and majors) regarding it, kindly find below my remarks.

Minor Comments:

01- In the whole manuscript, authors are invited to put all the following terms in italic: Pseudomonas, Pseudomonas aeruginosa and P. aeruginosa.

02- In the Introduction section, Lines 84-85 "Following a holistic approach, this work aimed to explain the synthesis of the sider- 84 ophore pyoverdine (PVD), a virulence factor in P. aeruginosa". Authors are invited to add the following reference: A Review of Pseudomonas aeruginosa Metallophores: Pyoverdine, Pyochelin and Pseudopaline.

03- In the Introduction section, Lines 107-108, "Most PVD biosynthesis is carried out in the bacterial cytosol by four non-ribosomal 107 peptide synthase (NRPS) proteins encoded by the pvdL, pvdI, pvdJ, and pvdD genes". Authors are invited to add the following reference: A Review of Pseudomonas aeruginosa Metallophores: Pyoverdine, Pyochelin and Pseudopaline.

Major Comments:

01- In the Results section, Authors are invited to remove or to merge some figures since they are numerous, they can move some of these figures as supplementary figures.

02- In the Results section, some figures are very large, authors are invited to minimize it.

03- In the Discussion section, authors are invited to talk about the idea that this study can be a starting point for the comprehension of the biosynthesis of other siderophore in Pseudomonas aeruginosa such as "Pyochelin" and the newly discovered narrow spectrum metallophore that acts as a virulent factor "Pseudopaline" and they can use the following articles as references: 

Ref 1: Pseudomonas aeruginosa zinc uptake in chelating environment is primarily mediated by the metallophore pseudopaline.

Ref 2: Chelating mechanisms of transition metals by bacterial metallophores “pseudopaline and staphylopine”: A quantum chemical assessment.

Best Regards,

Author Response

Dear

METABOLITES Reviewer 3

Section: Bioinformatics and Data Analysis

We are very grateful for the review process carried out on the manuscript entitled: A Holistic Approach from Systems Biology Reveals the Direct Influence of the Quorum-Sensing Phenomenon on Pseudomonas aeruginosa Metabolism to Pyoverdine Biosynthesis - METABOLITES- 2326047.  Thank you so much for your comments and for considering that our work is well written in English, well presented and it will be very attractive for readers. Following your suggestions, we addressed the minor and major changes to the document. These changes and responses to each point are described here:

Please check the responses in the document.

We would like to thank the referee for evaluating our manuscript. We have tried to address all the reviewers’ concerns in a proper way and believe that our manuscript has been significantly improved.

On behalf of my coauthors,

Diana Carolina Clavijo B. PhDc

Corresponding author.

Round 2

Reviewer 1 Report

A Holistic Approach from Systems Biology Reveals the Direct Influence of the Quorum-Sensing Phenomenon on Pseudomonas aeruginosa Metabolism to Pyoverdine Biosynthesis. Diana Carolina Clavijo-Buriticá, Catalina Arévalo-Ferro, and Andrés Fernando González Barrios.

The revised manuscript presents only minor changes. Although the authors answered all the comments raised by this reviewer, some responses were very superficial.

The two models derived in this work were interesting, however there are aspects related to their combination that are not well explained or not explained. Also, some of the assumptions are not valid. On the other hand, the experimental data used to validate the model prediction did not provide significant evidence.

In the conclusion section the authors say that “the DFBA approximation applied to the CCBM1146 model evidenced its high predictive capability for the behavioral pattern of an in vitro culture of P. aeruginosa” although this was not demonstrated. They also said that “it was possible to infer the influence of the QS phenomenon on the expression of different metabolic phenotypes of P. aeruginosa as a function of QS signal intensity” not explaining which metabolic phenotypes were considered and how was this inferred.

The work should not be considered for publication in its present form.

Other comments are:

486-488: What is the meaning of the saturation point? Results in Figure 4 showed that PVD concentration initially increases and then decreased; if the mass balance equation for this compound is similar to eq (2), then the increase is explained by a rate of synthesis higher that the rate of degradation, however after the concentration of PVD increases then the rate of degradation also increases and according to the results, this rate is higher than the rate of synthesis.

The results obtained with this model depend on the initial conditions; a sensitive analysis should be done to test this effect unless this information is available somewhere.

497-499: Model predictions need a deeper analysis; were the results at the steady state dependent on the concentration of E-PQS? Cell metabolism is generally considered be at steady state, were the results in agreement with experimental data? Was the transient behavior shown in Figure 4 representative of the in vivo process?

571-575: The fact that only 352 of the 1123 reaction carried flux is not related to the addition of PVD to the biomass reaction. This is due to the structure of the biomass equation.

576: If PVD is a secondary metabolite then a different objective function should be used. Secondary metabolites are synthesized not growth associated.

580: A justification of the DFBA simulation should be made. What do the authors need to demonstrate with these results?

Author Response

Dear

METABOLITES Reviewer 1

Section: Bioinformatics and Data Analysis

We are very grateful for the review process carried out on the manuscript entitled: A Holistic Approach from Systems Biology Reveals the Direct Influence of the Quorum-Sensing Phenomenon on Pseudomonas aeruginosa Metabolism to Pyoverdine Biosynthesis - METABOLITES- 2326047.  Thank you so much for your comments in the round 2. Following your suggestions, we addressed the minor and major changes to the document. These changes and responses to each point are described in the attachment.

 Please check the responses in the document.

 We would like to thank the referee for evaluating our manuscript. We have tried to address all the reviewers’ concerns in a proper way and believe that our manuscript has been significantly improved.

 On behalf of my coauthors,

 Diana Carolina Clavijo B. PhDc

Corresponding author.

Reviewer 3 Report

Dear Authors,

The resubmitted version of your work has been reviewed,

Thank you for all the corrections you did, this version is better than the previous one,

Best Regards,

Author Response

Dear

METABOLITES Reviewer 3

Section: Bioinformatics and Data Analysis

I hope this letter finds you well. I am writing to you regarding our manuscript entitled: "A Holistic Approach from Systems Biology Reveals the Direct Influence of the Quorum-Sensing Phenomenon on Pseudomonas aeruginosa Metabolism to Pyoverdine Biosynthesis - METABOLITES- 2326047" which was submitted to Metabolites for consideration. In addition, I am writing to express my gratitude for the opportunity to revise our manuscript based on your valuable feedback during the process.

Thank you so much for your comments and for considering that our work is complete and deserves to be published in Metabolites Journal.  

 On behalf of my coauthors,

Diana Carolina Clavijo B. PhDc

Corresponding author.

Round 3

Reviewer 1 Report

A Holistic Approach from Systems Biology Reveals the Direct Influence of the Quorum-Sensing Phenomenon on Pseudomonas aeruginosa Metabolism to Pyoverdine Biosynthesis.

Diana Carolina Clavijo-Buriticá, Catalina Arévalo-Ferro, and Andrés Fernando González Barrios

The second version of the manuscript included most of the suggestions and comments given to the authors by this reviewer. I believe the manuscript could be considered for publication in its present form.